# Combining Human Genetics of Multiple Sclerosis with Oxidative Stress Phenotype for Drug Repositioning

**DOI:** 10.3390/pharmaceutics13122064

**Published:** 2021-12-02

**Authors:** Stefania Olla, Maristella Steri, Alessia Formato, Michael B. Whalen, Silvia Corbisiero, Cristina Agresti

**Affiliations:** 1Istituto di Ricerca Genetica e Biomedica, Consiglio Nazionale delle Ricerche, 09042 Monserrato, Italy; maristella.steri@irgb.cnr.it; 2Department of Neuroscience, Istituto Superiore di Sanità, 00161 Rome, Italy; alessia.formato@iss.it (A.F.); silvia.corbisiero@iss.it (S.C.); 3Istituto di Biofisica, Consiglio Nazionale delle Ricerche (CNR), 38123 Trento, Italy; michaelbernard.whalen@ibf.cnr.it

**Keywords:** GWAS, multiple sclerosis, oxidative stress, repurposing, ADME-Tox

## Abstract

In multiple sclerosis (MS), oxidative stress (OS) is implicated in the neurodegenerative processes that occur from the beginning of the disease. Unchecked OS initiates a vicious circle caused by its crosstalk with inflammation, leading to demyelination, axonal damage and neuronal loss. The failure of MS antioxidant therapies relying on the use of endogenous and natural compounds drives the application of novel approaches to assess target relevance to the disease prior to preclinical testing of new drug candidates. To identify drugs that can act as regulators of intracellular oxidative homeostasis, we applied an in silico approach that links genome-wide MS associations and molecular quantitative trait loci (QTLs) to proteins of the OS pathway. We found 10 drugs with both central nervous system and oral bioavailability, targeting five out of the 21 top-scoring hits, including arginine methyltransferase (CARM1), which was first linked to MS. In particular, the direction of brain expression QTLs for CARM1 and protein kinase MAPK1 enabled us to select BIIB021 and PEITC drugs with the required target modulation. Our study highlights OS-related molecules regulated by functional MS variants that could be targeted by existing drugs as a supplement to the approved disease-modifying treatments.

## 1. Introduction

Multiple sclerosis (MS) is the most common chronic inflammatory and progressively disabling disease of the central nervous system (CNS), affecting young adults and leading to demyelination and neuronal degeneration [1]. It is found worldwide, with the highest prevalence (>100 cases per hundred thousand) in the populations of Western Europe, North America and Australasia, with considerably lower prevalence (<30 cases per hundred thousand) in populations that live nearer to the equator [2]. MS is likely the result of an interaction between genetic and environmental factors, but its etiology remains unknown. Although approved immunomodulatory therapies are effective in the early stages of the disease, they have little or no benefit in terms of preventing the transition to a more steadily progressive phase, characterized by accumulation of neuronal injury and loss. Thus, the search for agents that slow neurodegeneration and disability progression in MS is urgent.

Neuroinflammation is recognized as a key player in MS pathogenesis. It is present in all stages of the disease and involves adaptive and innate immune responses. Histopathological studies of MS indicate that demyelination and neurodegeneration are associated with the production of inflammatory molecules by both blood-derived immune cells recruited to the CNS and activated resident microglia [3]. Prolonged or chronic generation of cytokines, chemokines, reactive oxygen species (ROS) and reactive nitrogen species (RNS) creates a self-perpetuating loop that provokes CNS damage and is considered to play a key role in the onset and progression of the disease [4]. 

ROS and RNS, including superoxide ions, hydrogen peroxide, nitric oxide and peroxynitrite, are generated by NADPH oxidase and nitric oxide synthase during normal cellular metabolism. However, these molecules are deleterious if overproduced because they can damage lipids, proteins and nucleic acids, eventually leading to cell death. Significant evidence indicates that the sustained inflammatory phase of MS creates an imbalance between ROS/RNS generation and the antioxidant defense systems, causing oxidative/nitrosative stress which has a role in CNS tissue damage [5]. Antioxidant defense is normally achieved with enzymes, such as superoxide dismutase, catalases and peroxiredoxins, as well as systems of antioxidant production, like the thioredoxin and glutathione systems. In addition, reactive species directly interact with critical signaling molecules, such as the transcription factors nuclear factor-erythroid 2-related factor 2 (Nrf2) and nuclear factor κB (NfkB) and mitogen activated kinases (MAPK) [6,7,8] which regulate antioxidant gene expression and cell survival. A recent gene expression study of MS brain areas adjacent to perivascular inflammatory cell infiltrates showed a significant induction of antioxidant genes in actively demyelinating and chronically active white matter lesions as part of a counter-regulatory response aimed at containing inflammation and limiting tissue damage [9]. Hence, the identification of drugs able to effectively support the maintenance of redox homeostasis represents a rational approach to limit MS-associated neurodegenerative processes.

Among current MS drugs, only dimethyl fumarate has been linked to the induction of antioxidant pathways, specifically through direct activation of Nrf2, a transcription factor with a crucial role in the regulation of the antioxidant defense response [10]. In addition, the clinical efficacy of natalizumab and fingolimod could in part be explained by their ability to increase antioxidant molecules and reduce oxidative stress (OS) biomarkers in MS patients [11,12], even though the mechanism responsible for these effects has not yet been established. Nevertheless, most complementary antioxidant therapies relying on endogenous and natural compounds have been previously investigated without overcoming MS clinical evaluation [13]. A possible explanation of this oversight is that the rationale behind the use of small molecules acting as scavengers was based on misconceptions linked to an incomplete understanding of antioxidant defense processes during disease development [14]. Hence, novel approaches should be used to assess the disease relevance of antioxidant targets prior to preclinical testing of new drug candidates.

It is now widely accepted that the selection of targets based on genetics significantly increases the success rates of clinical development programs [15,16]. The idea is to identify targets involved in disease processes that can be therapeutically modulated [17,18]. Over the past fifteen years, genome-wide association studies (GWAS), in increasingly larger sample sets, have succeeded in identifying more than 200 susceptibility loci for MS outside the major histocompatibility complex (MHC) [19]. In parallel, new functional genomic techniques assessing molecular quantitative trait loci (QTLs), such as chromatin interactions, protein level and gene expression regulation, have proven to be useful for the systematic identification of genes through which trait-association variants act, improving the clinical impact of GWAS [20]. Computational searches for existing drugs that modulate the molecular targets identified by genetic studies offer the advantage of repositioning, reducing the costs and timescales of drug development. In addition, in silico approaches are currently being used for the prediction of physicochemical properties, such as the blood–brain barrier (BBB) permeability and oral bioavailability of drugs, further reducing the risk of failure [21]. 

Here, we designed and applied an integrated approach that combines MS GWAS, molecular QTLs and in silico techniques of drug discovery, providing support for single drug candidates known to act as modulators of genes and/or gene products that are linked to OS pathways (Figure 1). 

## 2. Materials and Methods

### 2.1. Data Collection

MS GWAS summary statistics were extracted from the GWAS Catalog [22,23]. The selected genetic variants represent the most associated signal (top variant) in each genomic region (locus) given a significance threshold of *p*-value < 1 × 10^−5^. All variants have been annotated by their rsID in dbSNP154, when available, or by chromosome and genomic positions encoded in the Genome Assembly GRCh38/hg38. To assign the most reliable gene target to each associated variant, molecular QTLs were searched for each hit in a large manually curated QTL resource, the LinDA browser [24,25]. Data from protein QTLs (pQTLs), expression QTLs (eQTLs), splicing QTLs (sQTLs), polyadenylation QTLs (polyQTLs) and methylation QTLs (mQTLs) were collected. The genomic positions in the LinDA browser being encoded in the Genome Assembly GRCh37/hg19, genomic coordinates were converted from GRCh38/hg38 to GRCh37/hg19 using the LitfOver tool in the UCSC Genome Browser [26,27].

Top variants were searched for molecular QTLs, including all variants showing a linkage disequilibrium (LD) r2 > 0.7 with the top variants (proxies) in the European population. LD was calculated using the --ld option in the plink v.1.9 software [28,29] on data from the 1000 Genome Project reference panel [30].

The functional role of each tested variant was further evaluated by the Variant Effect Predictor (VEP) tool [31], and missense or more deleterious variants with a deleteriousness score (combined annotation dependent depletion, CADD-Phred) > 15 were prioritized [32]. 

Genes regulated at the RNA or protein level by a hit variant (or by a variant in strong LD with a hit variant) or tagged by a functionally relevant variant were flagged as “gene targets”.

The direction of the effect of each disease risk variant on the target product was calculated to establish the direction of the gene target modulation by therapy. To this end, the disease risk alleles available from the GWAS Catalog were coupled with the molecular QTLs alleles by applying the Plink --ld option to the European ancestry genotypes encoded in the 1000 Genome Project reference panel [30]. The direction of the effect of the disease risk allele on the molecular QTL was thus indicated as positive if the coupled molecular QTL allele showed a positive effect, and negative otherwise.

In parallel, 22 pathways related to OS were identified by Reactome [33,34], and all proteins belonging to the pathways were extracted. 

Genes and/or proteins obtained by the overlapping between the MS-related genes and the OS-related proteins were recorded as “targets”.

For each target, a prioritization score was defined by leveraging the gene-level information derived from GWAS and from LD. In particular, for each target, all top variants, together with their molecular QTL proxies pointing to the same gene, were collected. A score was attributed to the target for each of the following criteria met by at least one collected variant:

Top hit significantly associated with MS (*p*-value < 5 × 10^−8^: score = 5, if lying in the MHC region (chr6:27–33 mb in GRCh37): score = 2);

Top hit having a high effect on the disease compared to all top hit effects (odds ratio, OR > 1.2), with a decreasing score depending on the LD with gene-level molecular QTLs (LD ≥ 0.99: score = 4; LD range (0.95–0.99): score = 3; LD range (0.90–0.95): score = 2; LD range (0.80–0.90): score = 1);

eQTL available (score = 10; if the eQTL acts in the brain: additional score = 5);

LD level between the top hit and the eQTL (LD ≥ 0.99: score = 5; LD range (0.95–0.99): score = 3; LD range (0.90–0.95): score = 2; LD range (0.80–0.90): score = 1); 

QTL (except eQTL) with LD ≥ 0.99 with top hit: score = 3.

An overall score was calculated as the sum of the partial scores and the top 25% targets were then prioritized. 

### 2.2. g:Profiler Analysis

To perform functional enrichment analysis, g:Profiler e94_eg41_p11_9f195a1 was used [35,36]. The parameters for the enrichment analysis were as follows. A specific organism was chosen: *H. sapiens* (human). Gene Ontology (GO) analyses, GO molecular function (GO:MF), GO cellular component (GO:CC) and GO biological process (GO:BP) were carried out sequentially. The biological pathways used were the Kyoto encyclopedia of genes and genomes (KEGG), Reactome (REAC) and WikiPathways (WP) databases. The protein databases used were the Human Protein Atlas and CORUM databases. The statistical domain scope was used only for annotated genes. The significance threshold was the g:SCS threshold. The user threshold was 0.05.

### 2.3. Drug Searching 

Four different databases, OpenTarget [37,38], SuperTarget [39], DrugBank [40,41] and DGIdb [42,43] were used to search for drugs related to the targets of interest.

### 2.4. In Silico Prediction of Physicochemical Properties of Drugs

The Absorption, Distribution, Metabolism, Excretion and Toxicity (ADME-Tox) profile of the investigated compounds was predicted using the Schrodinger QikProp tool (Small-Molecule Drug Discovery Suite 2021–1, Schrodinger, LLC, New York, NY, USA). QikProp uses several indicators to estimate the activity in the CNS and thus also the ability of a compound to cross the BBB. The three most important are: (i) LogBB, which represents the blood–brain partition coefficient; (ii) the Madin–Darby dog kidney cell model (apparent MDCK permeability), which estimates the penetration of the substances through a layer of these cells, measured in nm/sec; (iii) the predictor of activity in the CNS. The indicators used to evaluate oral absorption include Human Oral Absorption, Percent Human Absorption and apparent Caco2 permeability, Caco2 being a human colon carcinoma cell line used to predict human intestinal permeability and to investigate drug efflux.

## 3. Results

### 3.1. Genetic-Driven Identification of Targets Linked to Oxidative Pathways in MS 

We systematically collected GWAS data for MS from the GWAS Catalog (Methods), identifying 698 different genetic variants (hits; Appendix A, Appendix A). We then examined molecular QTLs to identify gene targets by searching each hit or its proxies (with r2 > 0.7) in the LinDA browser (Appendix A). This LD-based searching strategy allowed us to maximize the information collected, considering that differences in the genetic map and/or in the sample size used in each study (both on disease and molecular QTLs) could lead to the identification of different genetic variants representing the same genetic signal. In addition, we evaluated the functional role of each tested variant by VEP, focusing on missense or more deleterious variants with a CADD-Phred score >15 (Appendix A). Thus, each gene regulated at RNA or protein level by a hit variant or tagged by a functionally relevant variant, excluding MHC genes, was recorded for a total of 2,085 unique gene targets (Appendix A). In parallel, we extracted the proteins encoded by 931 unique targets included in 22 OS-related pathways from the Reactome database [44] (Appendix A). The overlap between the 2,085 MS-related gene targets and the 931 OS-related proteins led to the identification of 85 shared targets (Appendix A), including KEAP1 and HDAC1, which are both known to be modulated by drugs currently in use for MS (dimethyl fumarate and fingolimod, respectively). Among the 85 targets, 18 are supported by molecular QTLs in the brain (ASF1A, ATP6V1G2, BBC3, BCL2L11, CAPN1, CARM1, CHAC1, CRTC3, CSNK2B, DNM2, FOXO3, HSPA1L, KEAP1, MAPK1, NUP85, POM121C, PSMB9 and TRMT112). In addition, for each variant whose risk allele effect on the gene product was available in the brain, we were able to establish the direction of action (up or down) on the transcript/protein level and, consequently, to choose drugs with the proper mode of modulation: inhibition or activation (Appendix A). In particular, 10 targets were regulated by MS risk variants at some level in the brain, and among them we observed increased expression levels for seven targets (ASF1A, CAPN1, CARM1, CHAC1, NUP85, POM121C and TRMT112) and decreased levels for three targets (BBC3, MAPK1 and PSMB9). 

### 3.2. Functional Enrichment Analysis of the Identified Targets 

To obtain the enrichment information for the 85 candidate targets showing QTLs, g:Profiler analysis was performed [36]. The default analysis implemented in g:Profiler searches for pathways whose genes are significantly enriched (i.e., over-represented) in the target list of interest and compares them to all genes in the genome. Among the most significant pathways detected by REAC, “cellular response to stress” (*p*-value = 4.2 48 × 10^−36^) and “cellular responses to external stimuli” (*p*-value = 1. 326 × 10^−35^) have been pointed out, consistent with OS being the investigated disease phenotype (Figure 2). “Proteasome” (*p*-value = 8. 289 × 10^−7^) and “proteasome degradation” (*p*-value = 9. 576 × 10^−7^) have been identified as the most represented pathways by KEGG and WP, respectively. The three most significant cellular functions outlined by GO were “transcription factor binding” (GO:MF, 1. 942 × 10^−6^), “cellular response to stress” (GO:BP, 8. 022 × 10^−19^) and “cytosol” (GO:CC, 1. 303 × 10^−19^). Appendix A gives details of all the individual targets involved in the described analyses. 

### 3.3. Target Prioritization and Drug Search

To prioritize the 85 selected targets, we assigned to each of them a genetic-based score which considers the strength of association (variant effect magnitude and significance) with the disease, the presence of QTLs regulating the gene target at protein expression level, particularly in the brain, and the extent of LD supporting all the molecular information.

Based on the score distribution, we then fixed a threshold of score ≥20, which corresponds to the top 25% of the OS-related targets (Appendix A). We prioritized 21 targets (Appendix A), including seven targets regulated by eQTLs in the brain for which we established the required direction of modulation (TRMT112, CAPN1, ASF1A, NUP85 and CARM1, suggested to be inhibited, and BBC3 and MAPK1, suggested to be activated).

In four different databases, we searched for modulators of the 21 top-ranking targets (Appendix A), selecting only: (i) drugs approved or in clinical trials; (ii) drugs known to act directly on the specific target or as transcriptional target modulators based on established criteria (DGIdb interaction score >0.50 and published data on experimental validation); (iii) drugs having a mode of action consistent with the direction of the eQTL for the risk allele in the brain, if present. This analysis identified 35 modulators of six out of the 21 top targets (MAPK1, MAPK3, CARM1, CDK4, STAT3 and FOS), with a substantial number of drugs for each target, except for CARM1, which had only one. To increase the modulators of CARM1 and to investigate the druggability of the remaining top-ranking targets, we also looked for experimental drug trials, finding five CARM1 inhibitors and 11 compounds for two additional targets (NR1D1 and CAPN1). In addition, the presence of at least one modulator on Pharos makes the targets ASF1A, HVCN1 and YWHAQ druggable [45,46]. We then compiled a final list of 50 compounds for the next selection phase.

### 3.4. Pharmacokinetic Prioritization of the Selected Drugs 

By QikProp, the ADME-Tox properties of 35 repurposable drugs and 15 experimental compounds associated with the eight selected targets were predicted (Appendix A). Among the selection criteria, we prioritized the expected penetration into the CNS and the oral bioavailability, which are essential for maintaining drug function and potency towards the respective targets. In addition, physicochemical descriptors and other general properties related to good overall pharmacokinetics and metabolism profiles were considered. In detail, we selected compounds having (i) a value ≥0 for predicted CNS activity; (ii) medium–good values of logBB and MDCK apparent permeability; (iii) high values of human oral absorption and percent human oral absorption; (iv) medium–good values of Caco2 apparent permeability (Appendix A). Overall, this analysis identified 10 repurposable drugs (Table 1) and seven experimental compounds. The selected drugs include: (i) the CARM1 inhibitor BIIB021 in clinical trial for breast and gastrointestinal tumors; (ii) the MAPK1 activator PEITC in clinical trial for lung and oral cancer; (iii) four CDK4 inhibitors, ABEMACICLIB approved for breast cancer, ALVOCIDIB, MILCICLIB and PHA-793887 in clinical trials for several tumors; (iv) three STAT3 modulators, ERLOTINIB approved for lung cancer, ENMD1198 and ATIPRIMOD in trial for neuroendocrine cancer and multiple myeloma; (v) PILOCARPINE approved for the treatment of presbyopia as an inducer of FOS expression. Some of the drugs that are presented in Table 1 do not directly modulate the identified targets but may act through indirect mechanisms. The MAPK1-3 inhibitors, MK8353 and LY3214996, were removed from the list since they have a mechanism of modulation not consistent with the direction of eQTLs that we identified for MAPK1 in the brain. The seven experimental compounds that exceeded the pharmacokinetics investigation comprise three CARM1 inhibitors (MS049, MS023, TP064) and four NR1D1 modulators (agonists GSK4112, SR9009 and SR9011 and antagonist SR8278) (Appendix A).

## 4. Discussion

Advanced genetic analysis in MS has identified variants that clearly influence gene expression of CNS-resident immune cells [19], highlighting potential functional consequences for dysregulation of genes involved in the generation of inflammatory and oxidative mediators that trigger neurodegenerative processes. Our purpose was to link genome-wide MS associations and the correlated molecular QTLs to targets of OS pathways, improving the prediction of drug candidates that act as regulators of intracellular oxidative homeostasis. We selected 10 drugs already in use for cancer therapies that are specific for five out of the 21 top-scoring targets involved in the interplay between oxidation–apoptosis–autophagy–inflammation. Of these, MAPK1, STAT3, CDK4 and FOS targets have been indicated in previous MS GWAS [19,47,48,49], while the potential genetic link of CARM1 with MS is novel. However, drugs with CNS and oral bioavailability have not been predicted for any of these targets.

GWAS-associated genes have already resulted in candidate targets for drug discovery and repositioning in both complex and monogenic diseases [50]. Concerning MS, several studies have outlined the functional consequences of a set of disease variants [47] but these findings have not yet been translated into clinical practice. Moreover, the crosstalk between OS, neurodegeneration and neuroinflammation has a central role in the pathogenesis of MS [51].

In this study, we correlated MS susceptibility loci to OS pathways, finding those alleles (outside the MHC) that influence risk for this relevant disease phenotype. Notably, 85 shared targets were identified and ranked by assigning a score to each genetic outcome available. The reliability of our results is supported by the high score for KEAP1 and HDAC1, known targets of two drugs currently in use for MS, the antioxidant dimethyl fumarate and the immunomodulator fingolimod, respectively. As expected, our selected targets are linked with OS at different levels, in line with the dynamic outline of this process, which accounts for various interrelated events occurring in different cellular compartments. Our list includes: NCF4, a component of the NADPH oxidase system, and the proton channel HVCN1, which are involved in ROS generation [52]; MAPK1, MAPK3, STAT3 and FOS, inflammatory signaling molecules directly activated by ROS [53,54]; the arginine methyltransferase CARM1, a transcriptional co-activator known to regulate NFkB-dependent gene expression [55] and to be involved in cellular processes, such as autophagy, control of the cell cycle and differentiation [56]; the kinase CDK4, which promotes cellular growth by stimulation of mitochondrial biogenesis and concomitantly increases ROS generation [57]; the circadian gene NR1D1, which improves cellular bioenergetics and is regulated by OS and inflammation [58,59]. Interestingly, targets involved in complex regulatory mechanisms have recently attracted interest in the treatment of multifactorial diseases, such as neurodegenerative diseases, in which several biochemical events and molecular targets operate simultaneously [60].

Our approach of genetic-driven target identification is based on the integration of GWAS with eQTLs, especially those measured in brain tissues, to assess genes whose expression levels are modulated by non-coding disease-related variants [49]. The fact that 80% of the genetic variants identified by GWAS map in non-coding regions highlights the potential of functional genomic tools [50,61]. The use in this pipeline of different MS GWAS datasets, including those not containing complete whole-genome results, increased the number of potential candidate targets. Moreover, when the correspondence between a disease-risk variant and an eQTL allele has been derived, we were able to obtain important information about the direction of drug target modulation to be considered.

Query of public databases, combined with in silico pharmacokinetics, allowed for the selection of 10 drugs acting as modulators of five targets associated with oxidative pathways in MS. The direction of brain eQTLs for CARM1 and MAPK1 enabled us to identify two drugs with the required target modulation, prioritizing BIIB021 and PEITC over modulators of targets without the direction of their allelic effect. In particular, BIIB021 is a CARM1 and HSP90 inhibitor currently in clinical trials for treating hematopoietic malignancies and solid tumors (NCT01004081, NCT00618319 and NCT00344786) which easily crosses the BBB and can be administered orally. The drug mechanism responsible for CARM1 inhibition has not yet been defined, and there is the possibility that it acts indirectly via the inhibition of HSP90, which was identified as a CARM1 interactor (EP 3 208 615 B1). In addition, we also indicated highly selective inhibitors of CARM1, recently developed and tested in experimental models [62,63,64]. PEITC is an organosulfur bioactive compound, known as an MAPK1 activator, that is currently in trial for lung cancer and leukemia treatment (NCT00691132 and NCT00968461). Notably, the anti-inflammatory and antioxidant activity of PEITC has been extensively demonstrated in both in vitro and in vivo models. [65,66]. Of note, our in silico ADME analysis confirmed previous data on the BBB permeability of this drug [67]. 

Lack of data on the direction of the effects of MS risk variants in the modulation of STAT3, CDK4 and FOS in the brain does not allow the selection of drugs with adequate therapeutic modulation (activation or inhibition). Previous studies based on genetic variants and QTLs have suggested drugs for repurposing without exploiting the direction of effects [49,68], further supporting the potential relevance of our results.

In our study, we exclusively selected drugs that had passed clinical phase I and which therefore should be free of serious side effects regardless of their selectivity. Nevertheless, some drugs, including the CDKs inhibitor Alvocidib, present dose-dependent adverse effects that might be evaluated in the disease of interest by a risk–benefit analysis. As shown for CARM1, small molecules with a higher selectivity can be found among compounds active in preclinical studies but, by definition, these are not currently repurposable compounds.

The knowledge about targets relevant to OS in MS for which no approved modulators are currently available could be exploited in future drug discovery studies. Our search for experimental modulators of these targets led to the identification of NR1D1 agonists and antagonists [69], thus proving the druggability of an additional target.

A major limitation of our in silico approach concerns the finding that only about 22% of protein-coding genes are druggable [70], which is consistent with the low proportion of top-identified targets engaged by approved or in clinical trial drugs. A more stringent selection of genes strongly associated with disease may result in the loss of relevant targets showing small effect sizes [71]. In addition, the smaller number of QTLs assessed in the brain compared to other tissues and the lack of protein-QTLs significantly reduce the number of candidate genes to be matched with the selected disease phenotype. It should also be kept in mind that public databases for GWAS, drug targets and pathways make available data that are usually not uniform, often incomplete and frequently not up-to-date, and these represent important constraints for the achievement of a comprehensive analysis.

## 5. Conclusions

This study highlights the support of genetics in identifying targets which can potentially result in an unbalance of OS-related pathways in MS and existing drugs that can be repositioned to aim at these targets. We showed for the first time an increased expression of CARM1 genetically linked to MS. This finding agrees with the emerging dysregulation of methylation pathways in MS, which may impact immune and neurological processes [72]. Notably, several links between arginine methylation and neurodegenerative diseases, such as amyotrophic lateral sclerosis, Alzheimer’s and Huntington’s disease, have been established over the last few years [73]. However, preclinical studies will be necessary to validate the best drug candidates in cellular or animal models before their therapeutic application. A network pharmacology analysis could be helpful in identifying combinations of drugs targeting different unbalanced signaling pathways consistent with omics data integration and a multitarget drug development approach [74].

## Figures and Tables

**Figure 1 pharmaceutics-13-02064-f001:**
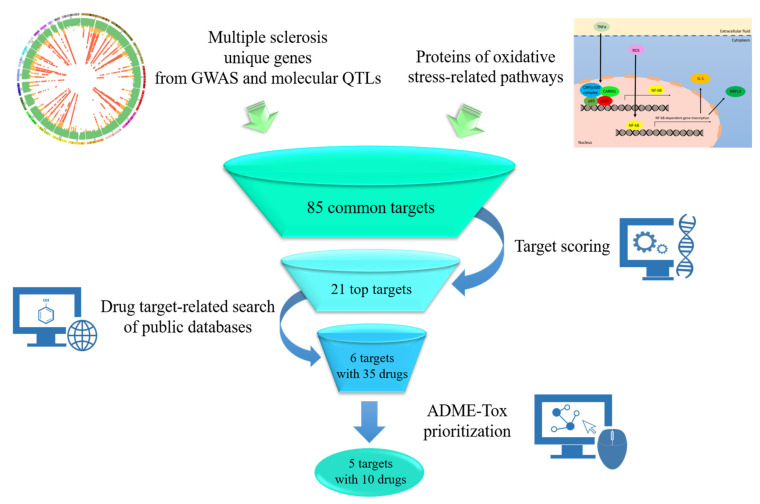
Schematic illustration of the in silico workflow. Multiple sclerosis (MS) genetic variants were collected from the Genome-Wide Association Studies (GWAS) Catalog and molecular Quantitative Trait Loci (QTLs) were exploited for each hit in the LinDA browser to identify gene targets. In parallel, all proteins from 22 oxidative stress-related pathways were retrieved from the Reactome database. The overlap of these data allowed for the identification of 85 common targets which were then prioritized through score assignment. Query of public drug databases for the 21 top targets enabled the selection of 35 drugs either already approved or in clinical trials that bind to six MS molecular targets. Absorption, Distribution, Metabolism, Excretion and Toxicity (ADME-Tox) selection highlighted 10 drugs with CNS localization and oral bioavailability for repurposing in MS.

**Figure 2 pharmaceutics-13-02064-f002:**
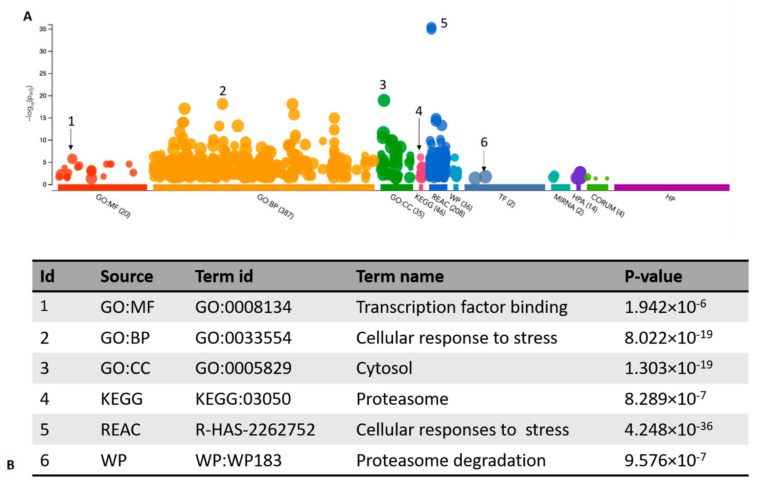
g:Profiler analysis of 85 targets. (**A**) Graphic representation of the results. (**B**) The most significant results for Gene Ontology (GO) and pathways enrichment were shown. GO molecular function (GO:MF); GO biological process (GO:BP); GO cellular component (GO:CC); Kyoto encyclopedia of genes and genomes (KEGG); Reactome (REAC); WikiPathways (WP).

**Table 1 pharmaceutics-13-02064-t001:** Repurposable candidates for oxidative-stress phenotype in MS. The table shows drug candidates with their mechanism of action and clinical trial status for each target. The queried databases are also reported.

Target	Drug	Mechanism of Action	Status *	Database
CARM1	BIIB021	HSP90 and CARM1 inhibitor	Phase II for breast cancer and gastrointestinal stromal tumors	DGIdb
MAPK1	PHENETHYL ISOTHIOCYANATE, (PEITC)	Bioactive compound activates ERK signal	Phase II lung cancer, tobacco use disorder and oral cancer	Super Target
CDK4	ABEMACICLIB	CDK4/6 inhibitor	Approved for breast cancer	DGIdb, DrugBank, OpenTarget
CDK4	ALVOCIDIB	CDKs inhibitor	Phase II for chronic lymphocytic leukemia; relapsed or refractory multiple myeloma; B-cell lymphoma; sarcoma; acute myeloid leukemia; prostate cancer; advanced ovarian epithelial cancer or primary peritoneal cancer; adenocarcinoma; kidney cancer; melanoma; endometrial cancer	DGIdb; DrugBank; OpenTarget
CDK4	MILCICLIB	CDKs inhibitor	Phase II for malignant thymoma and hepatocellular carcinoma	DGIdb; OpenTarget
CDK4	PHA-793887	CDKs inhibitor	Phase I for advanced- metastatic solid tumors	DGIdb
STAT3	ATIPRIMOD	Blocks STAT3 activation	Phase II for neuroendocrine cancer and multiple myeloma	DGIdb
STAT3	ENMD 1198	Mitosis inhibitors; tubulin modulators; STAT3 inhibitor	Phase I for advanced cancer	DrugBank
STAT3	ERLOTINIB	EGFR inhibitor; stimulated phosphorylation and activation of STAT3	Approved for lung and pancreatic cancer	SuperTarget; DGIdb
FOS	PILOCARPINE	Muscarinic receptor agonist- induced c-fos expression	Approved for the treatment of presbyopia	DGIdb

* Only the highest phase is shown.

## Data Availability

Not applicable.

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
