# Peer review of "Combining Human Genetics of Multiple Sclerosis with Oxidative Stress Phenotype for Drug Repositioning"

_pharmaceutics, 2021, doi:10.3390/pharmaceutics13122064_

Round 1
Reviewer 1 Report
In this paper, the authors made use of GWAS, pathway databases (focusing on oxidative stress) and other computational resources and approaches to prioritize drug targets and repositioning candidates for multiple sclerosis. The approach is clearly presented and logical, and the paper is well-written. A major limitation is that the approach employed is often ‘subjective’ (e.g. requires manual scoring of genes/variants); also, there is a lack of validation of the proposed approach, or the drugs highlighted. My further comments are listed below:
- 4 It is reasonable to employ GWAS data to prioritize drug targets. However, I feel the scoring a bit too ‘subjective’. For example, ‘eQTL available’ has a score =10 (max 15 if also expressed in brain), which is 2 or 3 times higher than a SNP with genome-wide significance. Scoring based on different LD is reasonable but again quite subjective. If possible, some kind of data-driven approach (e.g. machine learning) or more quantitative approaches (Eg considering the actual effect size /significance of the gene) should be considered. Also, the authors did not consider evidence from gene-based test (ie tests that aggregate evidence across multiple snps within the same gene).
Also, if a snp does not map to a gene and is not a known QTL, how will it be handled?
- Considering the overlap with oxidative-stress related genes is a reasonable approach. However, the exact direction of effects of genes contained in the database may not be clear, and relying on a single database (reactome) may not be ideal.
- After finding the top genes and drugs that may target these genes, the authors come up with repurposing candidates. One limitation is that one cannot easily/confidently determine the direction of the drugs from the present analytic framework. There is no explicit discussion on how the approach considers the direction of effect of the gene targets.
- It would also be nice to discuss briefly the practical usefulness of the candidate drugs, e.g. are they associated with severe side-effects?
Reviewer 2 Report
interesting, congrats
Author Response
We thank the reviewer for the positive feedback.
There are no criticisms or suggestions.
Reviewer 3 Report
pharmaceutics-1442358, Combining human genetics of multiple sclerosis with oxidative stress phenotype for drug repositioning
row 28, “is the most common chronic inflammatory“. How common? The authors should provide some incidence values to prove this statement.
The introduction focus mainly on reactive oxygen species, and almost ignores the inflammatory part. The authors should also describe this very important mechanism.
On row 65, describe also the main mechanism of both natalizumab and fingolimod
There are many other treatment possibilities for MS that are ignored by the authors and that should be at least presented. See: https://pubmed.ncbi.nlm.nih.gov/31643690/
The authors describe in the paper genes as targets. I think the proteins encoded by the genes are the real targets. The authors should be more cautious in their presentation of the methods and results when talking about proteins or genes because it is very confusing. The authors should present the names of the proteins targeted by drugs, and not of the genes. For example, it should be Histone-arginine methyltransferase, and not CARM1. The symbols for genes should be italicized.
The drugs presented in table 1 should be checked. As far as I know, Erlotinib is an EGFR inhibitors, and not of STAT3.
The authors should point out the weak points of the study and the limitations of the methods used. In my opinion, one major problem is that the proteins targets identified in the study are involved in complex mechanisms of regulation and often are interconnected.
Overall, the authors should be more objective with their data and their results. For example, the presentation of Alvociclib. See the description from the article “The application and prospect of CDK4/6 inhibitors in malignant solid tumors”. It states that: “Alvociclib, the first-generation CDK inhibitor, lacks specificity and blocks CDK1/2/4/6/7/9, causing serious adverse effects and limiting its clinical application”. The authors are ignoring all the side effects and problems with their candidates. They should correct that.
Round 2
Reviewer 3 Report
The authors responded to most of my comments. Still, most of their responses are adresed to the reviewer, and not to the readers. I advise the authors to include their comment also in the manuscript in the discussion section. The readers should be able to see the limitations and possible errors associated with this method.
Author Response
In this second revision of the manuscript, additional comments related to issues raised by the reviewer were included in results (lines 288-289) and discussion (lines 337-340 and 373-378) to better highlight the potential limitations of the proposed approach.
We thank the reviewer for the helpful suggestions.